# Identification of Transcription Factor Networks during Mouse Hindlimb Development

**DOI:** 10.3390/cells12010028

**Published:** 2022-12-21

**Authors:** Maochun Wang, Ruiyang Jiang, Guihua Tan, Zizheng Liu, Chen Tang, Rui Wu, Dongquan Shi

**Affiliations:** 1State Key Laboratory of Pharmaceutical Biotechnology, Department of Sports Medicine and Adult Reconstructive Surgery, Affiliated Drum Tower Hospital, Medical School, Nanjing University, 321 Zhongshan Road, Nanjing 210008, China; 2Nanjing Drum Tower Hospital Clinical College of Xuzhou Medical University, Xuzhou Medical University, Nanjing 210008, China; 3College of Biological Science and Medical Engineering, Donghua University, Shanghai 201620, China

**Keywords:** transcription factor, hindlimb development, WGCNA, regulatory network, Sox9, Twist1, chondrogenesis

## Abstract

Mammalian hindlimb development involves a variety of cells and the regulation of spatiotemporal molecular events, but regulatory networks of transcription factors contributing to hindlimb morphogenesis are not well understood. Here, we identified transcription factor networks during mouse hindlimb morphology establishment through transcriptome analysis. We used four stages of embryonic hindlimb transcription profiles acquired from the Gene Expression Omnibus database (GSE30138), including E10.5, E11.5, E12.5 and E13.5, to construct a gene network using Weighted Gene Co-expression Network Analysis (WGCNA), and defined seven stage-associated modules. After filtering 7625 hub genes, we further prioritized 555 transcription factors with AnimalTFDB3.0. Gene ontology enrichment showed that transcription factors of different modules were enriched in muscle tissue development, connective tissue development, embryonic organ development, skeletal system morphogenesis, pattern specification process and urogenital system development separately. Six regulatory networks were constructed with key transcription factors, which contribute to the development of different tissues. Knockdown of four transcription factors from regulatory networks, including Sox9, Twist1, Snai2 and Klf4, showed that the expression of limb-development-related genes was also inhibited, which indicated the crucial role of transcription factor networks in hindlimb development.

## 1. Introduction

Limbs are an important organ of vertebrates. They involvedcell proliferation, differentiation, migration and apoptosis during development [1]. Limb bud initiation is a sign of limb development. It contains cells of ectoderm and mesenchymal stem cells (MSC) of mesoderm, which differentiate into various tissues, including cartilage, muscle, vessel, tendon, ligament and bone [2,3]. Limb bud growth is regulated by a variety of developmental signaling pathways, in which FGF, BMP, HOX, SHH and WNT signal pathways can define three-dimensional structures and patterning [4,5]. It is thought that FGF controls limb proliferation and differentiation [1], and BMP family members regulate the bone formation process [6] and HOX family members are crucial for limb morphogenesis [5,7], while SHH and WNT pathways are necessary for limb axial patterning [8,9]. In humans, congenital limb malformations occur in about 1 in 500 neonates, and it is one of the common congenital defects [10]. This is caused by either genetic mutation or environmental teratogenic factors, which lead to limb abnormality [11]. However, spatiotemporal molecular events and crosstalk between different cells during limb establishment are not fully understood. Therefore, gaining an understanding of the molecular network during embryonic limb development will assist medical research and prenatal diagnosis of limb abnormalities.

Over the past decades, a large number of new technologies have emerged, including microarrays, RNA-seq and single cell sequencing, which provide the opportunity to comprehensively observe dynamic changes of gene expression during limb development [12,13,14]. Microarrays have helped to discover a series of novel candidate genes, which are associated with limb development in embryonic mice [12]. New specific markers of limb tendons are identified by bulk RNA-seq [15]. With the emergence of big data on sequencing, several algorithms have been developed to analyze genes that serve as signaling centers [16]. Weighted gene co-expression network analysis (WGCNA) is the most commonly used method to construct gene networks, and identify trait-associated modules, which enable investigators to identify hub genes [17]. The mechanism of the difference between hindlimb and forelimb development is rarely known. Several transcription factors (TFs) are specific contributers to hindlimb development, including Pitx1, Tbx4 [2,18,19]. Combining WGCNA with mouse hindlimb transcriptome may reveal more insights of hindlimb development. 

Here we used embryonic hindlimb transcriptome to identify the TF network by WGCNA, and we found several tissue-specific networks which could play important roles in the morphogenesis of hindlimb development (Figure 1A). Inhibition of four TFs (Sox9, Twist1, Snai2 and Klf4) from regulatory network was examined in ATDC5 cells, and development-related genes were also down regulated, which demonstrate the important roles of these TF networks.

## 2. Materials and Methods

### 2.1. Data Acquisition

Raw data was downloaded from the Gene Expression Omnibus (GEO) database. Hindlimb transcriptome data from 19 mice were collected from dataset GSE30138 [12], which was based on the Affymetrix Mouse Genome 430 2.0 Array. The collected hindlimb transcriptome data contained four different developmental stages (E10.5–E13.5). There were 45,101 probes in each array, and 20,507 unique genes were extracted. One gene symbol corresponded to multiple probe signal values, and the expression level of each gene was calculated as the average value of those probes. All transcriptome data were normalized by the Robust Multiarray Average (RMA) algorithm with affy (v 1.68.0) R package.

### 2.2. Weighted Gene Co-Expression Network Analysis

The gene co-expression network was constructed by WGCNA (v 1.69) R package [17]. Pair-wise Pearson correlation coefficients were calculated with the expression level of all genes and samples. Soft thresholding power β was set to 6 based on scale-free topology (R^2^ > 0.85) to create weighted adjacency matrix (WAM). Topological overlap matrix (TOM) and corresponding dissimilarity (1-TOM) were calculated by WAM. Genes were assembled into modules by hierarchical clustering, and further defined by the dynamic tree cut algorithm.

### 2.3. Identification of Significant Modules and Hub Genes

To identify significant modules associated with each stage of hindlimb development, correlations between module eigengenes (MEs) and each stage trait were assessed by the WGCNA R package. Stage-trait-associated modules were screened by *p* value < 0.001. Genes with high gene significance (GS > 0.2) and high intramodular connectivity (module membership [MM] > 0.8) in each module were defined as hub genes.

### 2.4. Construction of Transcription Factor Network

TFs of each stage-specific module were prioritized through hub genes by the AnimalTFDB3.0 database (http://bioinfo.life.hust.edu.cn/AnimalTFDB/#!/ (accessed on 13 January 2022)) [20]. Gene ontology of TFs was conducted by clusterProfiler (v 4.2.2) R package. TF network was constructed based on protein–protein interaction (PPI) of the STRING database and visualized by Cytoscape (v 3.9.1) software. The top 15 TFs of each network were obtained using cytoHubba plug-in according to the degree of nodes. PheWAS analysis was performed on a GWAS ATLAS (https://atlas.ctglab.nl/PheWAS (accessed on 25 March 2022)) [21]. Gene symbol was imported and visualized at the PheWAS plot with default parameters.

### 2.5. Validation in Mice Limb Tissue and ATDC5 Cells

All animal procedures were performed in the animal experiment center of Nanjing Drum Tower Hospital and approved by the Institutional Animal Care and Use Committee of the Nanjing Drum Tower Hospital, the Affiliated Hospital of Nanjing University Medical School (2020AE01102). 

Five TFs, including Mef2c, Six1, Smad3, Sox9 and Trp53, were verified by qRT-PCR. Hindlimb of embryonic mice from E10.5–E14.5 were harvested from pregnant mice. RNA was extracted by Trizol (Thermo Fisher Scientific, Waltham, MA, USA) and reverse-transcribed by HiScript III RT SuperMix for qPCR (Vazyme, Nanjing, China). qPCR was performed with ChamQ SYBR Color qPCR Master Mix (Vazyme, Nanjing, China) on LightCycler480 (Roche, Basel, Switzerland). Relative expression data were shown as mean ± SEM by GraphPad Prism 9. 

Knockdown of four TFs, including Sox9, Twist1, Snai2 and Klf4, were examined in ATDC5 cells, which is a mouse embryonic tumor cell line. Cells were seeded in a 12-well plate and cultured at 37 °C with 5% CO_2_. Two different siRNAs of four TFs were transfected with Lipofectamine 3000 (Thermo Fisher Scientific, Waltham, MA, USA), and scrambled siRNA was used as negative control. After transfection of 48 h, total RNA was extracted by Trizol (Thermo Fisher Scientific, Waltham, MA, USA) and reverse-transcribed by HiScript III RT SuperMix for qPCR (Vazyme, Nanjing, China). qPCR was performed with ChamQ SYBR Color qPCR Master Mix (Vazyme, Nanjing, China) on LightCycler480 (Roche, Basel, Switzerland). Relative expression data of four TFs and different markers were shown as the mean ± SEM by GraphPad Prism 9. A micromass culture of ATDC5 cells was carried out according to the protocol [22], and cells were visualized by 1% Alcian Blue (Sigma, Darmstadt, Germany).

## 3. Results

### 3.1. Quality Control of Embryonic Mice Hindlimb Transcriptome

A box plot showed that signals of raw data were uneven (Appendix A). After RMA normalization, the median of the signal was normalized (Appendix A). Cluster dendrogram and PCA analysis results showed that hindlimbs from the same stage clustered respectively (Figure 1B,C). Among them, E11.5 and E12.5 were at the transitional stage, and their transcriptional expression profiles were relatively similar. Whereas E10.5 was at the initial stage and E13.5 was at a later stage of limb morphogenesis, and their transcriptional expression profiles showed more differences. 

### 3.2. Weighted Gene Co-Expression Network Analysis

To identify regulatory genes during hindlimb development, a gene co-expression network was constructed by WGCNA. When the scale-free fit index was 0.9, the minimum soft-thresholding power was 6 (β = 6, Figure 2A, left), and the corresponding mean connectivity was about 2000 (Figure 2A, right), so β = 6 was selected as the optimal soft-thresholding power for subsequent analysis. All genes were imported in WGCNA, and a total of nine modules were identified with default setting parameters (Figure 2B). The size of module varied from 61 genes in modules green and yellow to 10,997 genes in module turquoise. Module-trait relationships showed that modules purple, brown, turquoise and yellow were significantly associated with initial stage E10.5, whereas modules green, blue, turquoise and magenta were significantly associated with later stage E13.5 (Figure 2C). Module membership and gene significance also showed a strong correlation in these modules (Appendix A).

### 3.3. Gene Expression Patterns in Different Modules

According to the expression patterns, genes could be divided into four categories, including early gene, late gene, intermediate gene and swing gene. The early genes were from module turquoise, and the expression was high at E10.5 which was the initiation of hindlimb bud, and the expression decreased gradually in the later stage of hindlimb development (Figure 3A). The late genes were from modules blue and brown, and the expression pattern was opposite of the early genes, which was highly expressed in the late stage of hindlimb morphogenesis (Figure 3B). The intermediate genes were from modules magenta and yellow, and the expression was greater at E11.5–12.5 in the middle stage, but lower in the early and late stages (Figure 3C). The swing genes were from modules green and purple, and the expression pattern was opposite that of intermediate genes, which displayed greater expression at E10.5 and E13.5 and lower expression at E11.5 and E12.5 (Figure 3D). Several genes, including Mef2c, Six1, Smad3, Sox9 and Trp53, were selected to demonstrate the reliability of microarray data and were verified in the hindlimbs of embryonic mice at different developmental stages. The results of qPCR proved that the microarray was reliable, and the expression trend was consistent in E10.5–E13.5 (Appendix A). According to eigengene adjacency heatmap, modules magenta and yellow, modules blue and brown, and modules green and purple showed strong correlations (Appendix A), which was also consistent with their gene expression patterns. 

### 3.4. Enrichment of Transcription Factor and Network Construction

There were 7625 hub-genes with high gene significance (GS > 0.2) and high intramodular connectivity (module membership [MM] > 0.8) (Appendix A). Based on gene ontology of molecular function, genes in modules blue, brown and green were mainly enriched in molecular binding, genes in module magenta were enriched in transcription repressor activity, genes in module purple were mainly enriched in transmembrane transporter activity, genes in module turquoise were mainly enriched in enzymatic activity, and genes in module yellow were mainly enriched in transcription coregulator activity (Appendix A).

Since TFs were crucial for gene regulation during hindlimb development, we further prioritized hub-TFs in each module through AnimalTFB3.0. Enrichment of biological process showed hub-TFs in different modules were enriched in different processes during hindlimb development (Figure 4A). TFs in module blue were related to muscle tissue development, TFs in module brown were related to connective tissue development and ossification, TFs in module green were related to semicircular canal development, TFs in module magenta were related to embryonic organ development and cell fate commitment, TFs in module purple were related to skeletal system morphogenesis, TFs in module turquoise were related to pattern specification process, and TFs in module yellow were related to urogenital system development (Figure 4A). Each of them showed correlation with different tissue development of hindlimbs.

In order to construct the molecular network of TFs, we imported the hub TFs of each stage-trait-related modules into the STRING database to identify their interactions. After the visualization of Cytoscape, top 15 TFs were prioritized by a cytoHubba plug-in according to the degree of nodes in the network (Figure 4B). These TF networks also showed agreement with the results of GO enrichment analysis. For example, module blue was mainly related to muscle tissue development, of which Mef2c was the myogenic marker, and module brown was associated with ossification, of which Runx2 was the phenotypic marker of osteoblast. In addition, PheWAS analysis of these key TFs from regulatory networks showed ESR1, ISL1, RUNX2, SIX1 and SMAD3 were associated with human skeletal phenotypes, including bone mineral density, height, osteoarthritis of hip or knee; FOXO1, MEF2C and TRP53 were associated with human metabolic phenotypes, including leg fat ratio, fat mass or percentage (Appendix A).

### 3.5. Inhibition of Key Transcription Factors Lead to Disruption of Limb Development Signals

In order to further verify the bioinformatic findings, we inhibited four TFs, including Sox9, Twist1, Snai2 and Klf4, and examined in ATDC5 cells. In contrast to negative control, chondrogenesis of ATDC5 cells were inhibited after knockdown of Sox9, Twist1, Snai2 and Klf4, respectively (Figure 5A,B). Knockdown of Twist1 and Snai2 had more impact on the staining area of ATDC5 cells than Sox9 and Klf4 (Figure 5B), which indicated the importance of Twist1 and Snai2 in chondrogenesis. Gli1, the effector of Hedgehog (Hh) signaling, decreased after knockdown of four TFs; Cdh5, a blood vessel marker of embryonic mouse limb development, also decreased after knockdown of four TFs; Jun, the first oncogenic transcription factor, was inhibited after knockdown of Sox9 and Klf4; Col1a1 and Col2a1, the important component of extracellular matrix, were also disrupted after knockdown of four TFs (Figure 5C). 

## 4. Discussion

Recent works have demonstrated that WGCNA is a useful way to identify hub-genes in tissue development [23,24,25]. There are no studies that combined WGCNA with mouse hindlimb transcriptome that we know of so far. Here, we integrated 19 mice hindlimb transcriptome, including four embryonic developmental stages, with WGCNA to identify TF networks. There were 555 hub-TFs in seven stage-associated modules, which consisted of six TF networks. The major gene enrichment of each network was related to different tissues of limb development, which could be a guiding role in tissue morphogenesis, such as bone, muscle and connective tissue.

Gene expression pattern of hindlimbs in mice was divided into four groups, including early gene, late gene, intermediate gene and swing gene (Figure 3). This was similar to the study of the temporal expression pattern in the developing mouse forelimb, which was also grouped into four clusters [12]. Our results showed three major expression patterns that can be further sub-grouped. The late genes from modules blue and brown were mainly highly expressed in E13.5, whereas genes from module brown also displayed E12.5 high expression. This indicated that genes in module brown functioned earlier than genes in module blue. The intermediate genes from modules magenta and yellow showed low expression at E10.5 and E13.5, but genes from module yellow showed greater expression than genes from module magenta at E13.5, which suggests that genes in module yellow may still have a functional role at E13.5. The general trend of swing genes from modules green and purple was the same, but there were some differences. Their expression levels were opposite at E10.5 and E13.5, which implies that they were functionally complementary to each other. The early genes from module turquoise showed high expression in E10.5, which was the early stage of hindlimb bud formation [14]. It contained the most genes, which was consistent with the results from transcriptomic analysis of mouse limb tendon development. The early stage had more up-regulated genes than the late stage during mouse limb tendon development [13]. In general, hindlimb gene expression patterns were more unique in the early (E10.5) and late stage (E13.5) of morphogenesis, while gene expression patterns in the transition stage (E11.5 and E12.5) displayed much higher similarity, which can also be demonstrated from the cluster dendrogram of transcriptome data (Figure 1B). Results of open-chromatin regions in E11.5 and E12.5 mouse limb using ATAC-seq indicates the similarity of molecular events from the transition stage [26].

TFs played important regulatory roles in the morphogenesis of limb development [27,28], and they were very important in the differentiation of progenitor cells in different tissues, including bone, muscle, connective tissue, etc. [19,29]. We constructed six major TF networks associated with hindlimb development (Figure 4). Of which, Pax7 (blue TF network) regulated muscle progenitor cell functions and was crucial for skeletal muscle development and regeneration [30]. Heterozygous knockout mice of Twist1 could cause limb malformations [31], and Twist1 was in the blue TF network. Runx2 (brown TF network) was essential for differentiation of osteoblast and chondrocyte during skeletal development [32]. Sox9 conditional knockout mice displayed abnormal limb stylopod morphology [33], this was consistent with the importance of Sox9 in the brown network. The Zeb1 (green TF network) homozygous mutation in mice exhibited abnormalities in the craniofacial region, limb, ribs, sternum and intervertebral discs [34]. Smad2 and Smad3 (magenta TF network) played a central role in chondrocyte proliferation and differentiation [35]. Mycn (turquoise TF network) was required for interdigital mesenchymal cells to trigger programmed cell death during limb pattern formation [36]. Shox2 (yellow TF network) regulated chondrogenesis and osteogenesis in the mouse proximal limb mesenchyme [37,38]. Scx, a limb tendon specific TF which played an important role in tendon cell differentiation [13], was in our yellow TF network. In summary, these results highlighted the importance of the TF networks in hindlimb development.

In addition, we verified several TFs in ATDC5 cells (Figure 5). Sox9 was important for limb chondrogenesis in the early stage [39]. Twist1 played a crucial role in mesoderm development, particularly in limb and craniofacial formation [31]. Klf4 and Snai2 were newly identified transcription factors, which were associated with hindlimb development. The knockdown results showed that disruption of these TFs inhibited the chondrogenesis of ATDC5 cells and suppressed transcriptional changes of limb-development-associated genes. We also found many TFs that were highly expressed in embryonic hindlimbs, but have not yet reported their functions during limb morphogenesis. In humans, PheWAS results showed that TFs in these networks were associated with various phenotypes, such as bone mineral density, height, weight and leg fat ratio (Appendix A), further indicating the importance of these networks for limb development. 

In summary, we combined WGCNA and mouse embryonic hindlimb transcriptional profiles and identified 7 stage-associated modules and 555 hub-TFs. Six regulatory networks were constructed with key TFs, which contributed to the development of different limb tissues. However, these genes need further verification to illustrate their potential roles during hindlimb morphogenesis. We hope that these results will help to supplement the research of limb development and the screening targets of human congenital limb defects.

## Figures and Tables

**Figure 1 cells-12-00028-f001:**
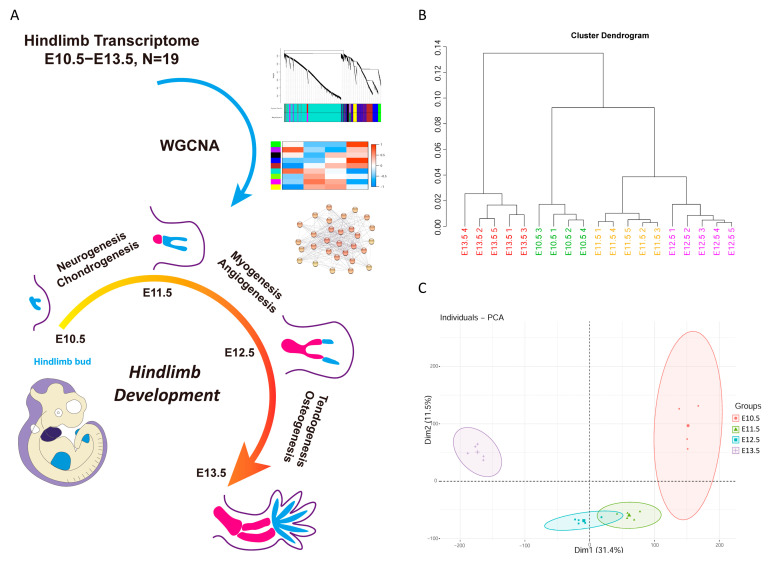
Study design and quality control of embryonic mice hindlimb transcriptome. (**A**) Workflow of hindlimb transcriptome and WGCNA analysis. Nineteen hindlimb transcriptome including four developmental stages from E10.5 to E13.5 were analyzed by WGCNA. Stage-associated modules were revealed and further transcription factor networks were constructed. (**B**) Cluster dendrogram of 19 hindlimb transcriptomes showed similarity of respective stage transcriptome. Light green, E10.5; yellow, E11.5; purple, E12.5; red, E13.5. (**C**) Principal component analysis of hindlimb transcriptome with four stages confirmed that different stages clustered separately. Circle, E10.5; triangle, E11.5; square, E12.5; cruciform, E13.5.

**Figure 2 cells-12-00028-f002:**
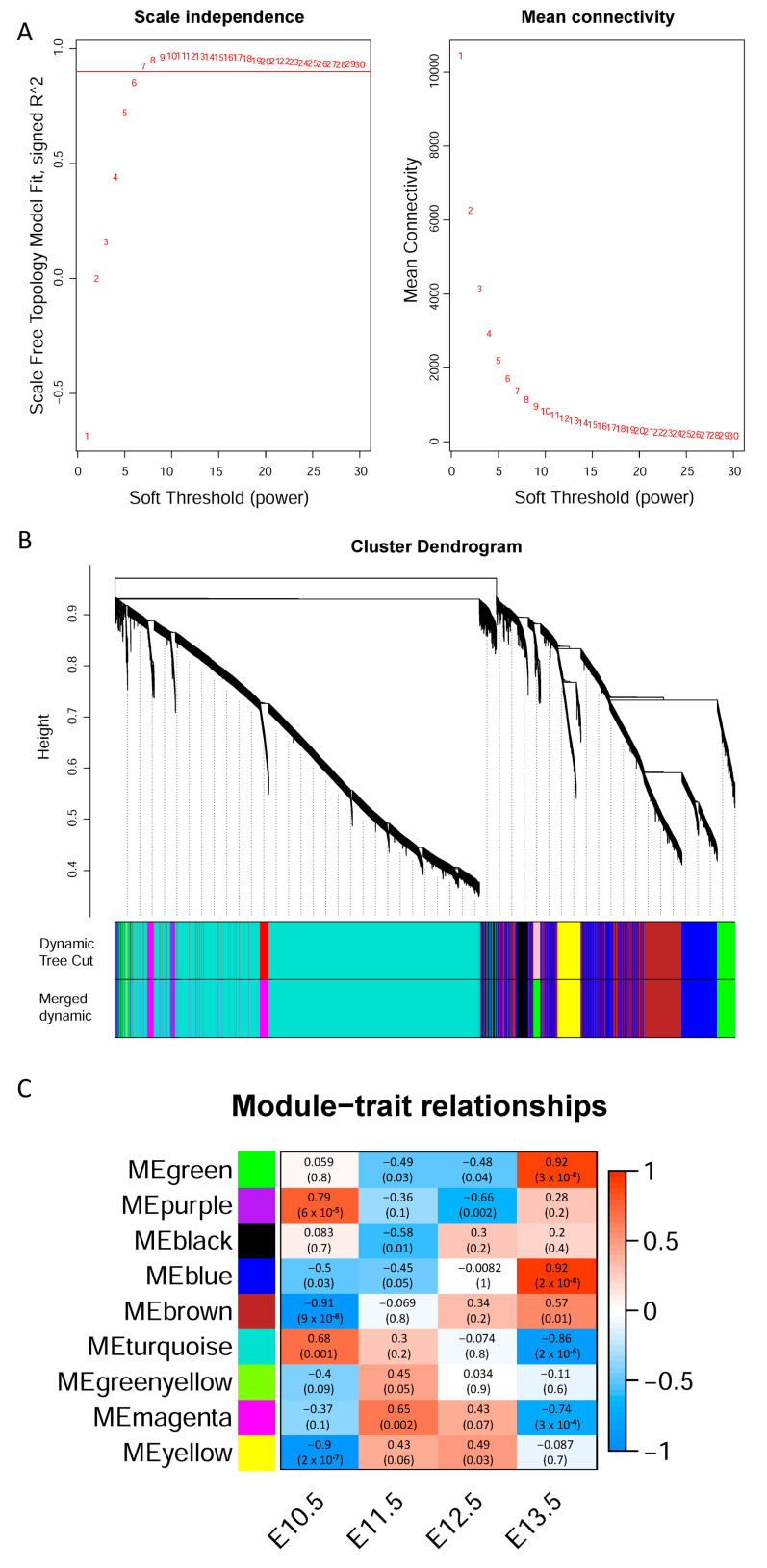
Weighted gene co-expression network analysis of hindlimb transcriptome. (**A**) Scale-free topology fit index R^2^ as a function of the soft thresholding power (β) plot (left). Mean connectivity as a function of the soft thresholding power (β) plot (right). (**B**) Network dendrogram of WGCNA by hierarchical clustering with dynamic tree cut and merged dynamic modules. Modules were divided according to height with dynamic tree-cut algorithm in the upper panel, and modules were merged according to their similar expression patterns with the merged dynamic algorithm in the lower panel. (**C**) Association of nine modules and four developmental stages. In each box, the upper value represents the association coefficient, and the lower value represents the significance.

**Figure 3 cells-12-00028-f003:**
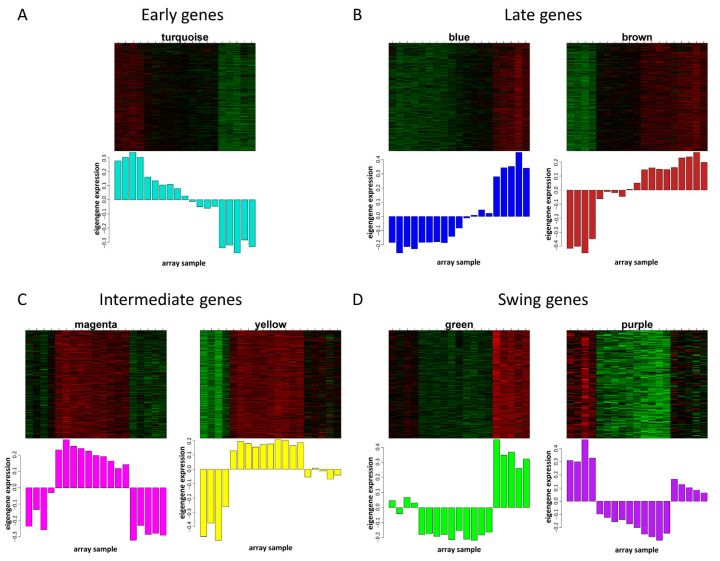
Gene expression patterns of different modules during mouse hindlimb development. Gene expression pattern of (**A**) early genes in module turquoise, (**B**) late genes in modules blue and brown, (**C**) intermediate genes in modules magenta and yellow, (**D**) swing genes in modules green and purple. Each figure above represents a heatmap of gene expression, and each figure below represents a histogram of gene expression across different samples.

**Figure 4 cells-12-00028-f004:**
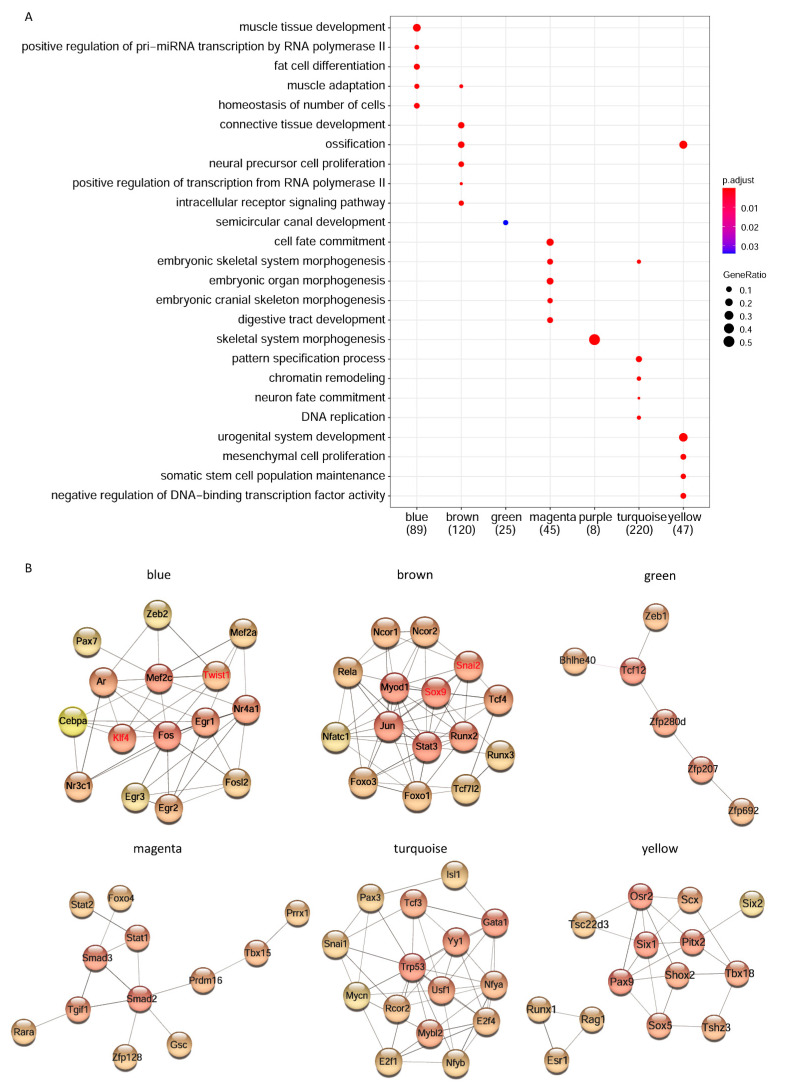
Construction of the transcription factor network during mouse hindlimb development. (**A**) Biological process enrichment of transcription factors in stage-associated modules. (**B**) Regulatory networks of transcription factors were driven from each module. Module purple was excluded as there were too few genes to construct a network. The redness of the color indicates more connectiveness of the transcription factor in the network. The names of four verified transcription factors, Sox9, Twist1, Klf4 and Snai2, are marked with red.

**Figure 5 cells-12-00028-f005:**
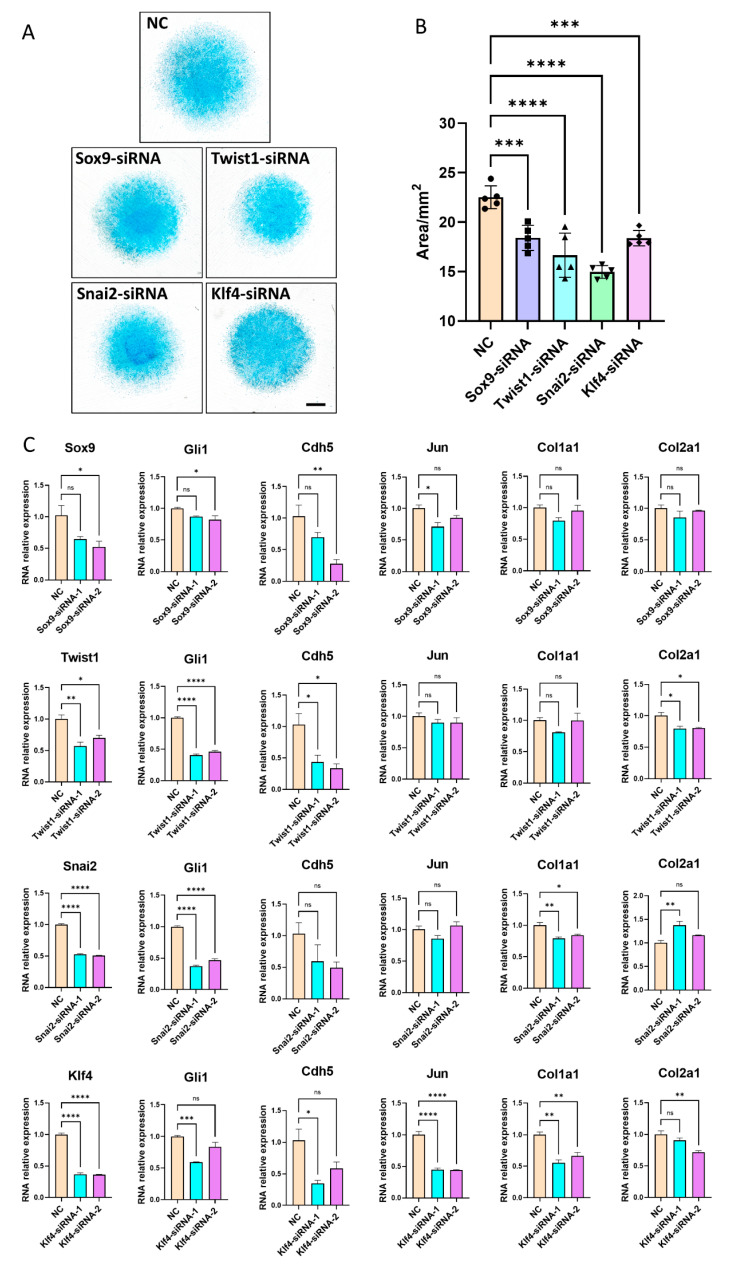
Inhibition of Sox9, Twist1, Snai2 and Klf4 in ATDC5 cells. (**A**) Alcian blue staining of ATDC5 cells after knockdown of transcription factor Sox9, Twist1, Snai2 and Klf4, respectively. The bar represents 1 mm. (**B**) Statistics of Alcian blue staining area. Data are presented as the mean ± SEM. (**C**) RNA relative expression of Sox9, Twist1, Snai2, Klf4, Gli1, Cdh5, Jun, Col1a1 and Col2a1 in ATDC5 cells after knockdown of four transcription factors, respectively. NC is negative control. Data are presented as the mean ± SEM. * *p* < 0.05; ** *p* < 0.01; *** *p* < 0.001; **** *p* < 0.0001; ns means no statistical significance.

## Data Availability

Raw data of dataset GSE30138 are available on GEO database (https://www.ncbi.nlm.nih.gov/geo/query/acc.cgi?acc=GSE30138), accessed on 11 September 2021; AnimalTFDB3.0 database (http://bioinfo.life.hust.edu.cn/AnimalTFDB/#!/), accessed on 13 January 2022; GWAS ATLAS (https://atlas.ctglab.nl/PheWAS), accessed on 25 March 2022.

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
