# Peer review of "Identification of Transcription Factor Networks during Mouse Hindlimb Development"

_cells, 2022, doi:10.3390/cells12010028_

Round 1

Reviewer 1 Report

In this manuscript, the authors compare gene expression among four stages of hindlimb development at E10.5, 11.5. 12.5 and 13.5 from limb bud appearance through osteogenesis using published Affymetrix mouse genome data in the GEO database. They identified nine modules and seven temporal patterns for gene expression. They then focused on transcription factors (TFs) in the datasets and identified biological processes associated with “hub” TFs, constructing six TF networks. Validation of four key TFs was accomplished by knock-out experiments in ATDC5 cells and subsequent quantitation of mRNAs. In the discussion, they note that their findings validate prior association of TF networks with specific functions such as chondrogenesis or limb pattern formation.

Overall, the findings are a minimal advancement in the field. The many tissues present in the hindlimb complicate any application of the results. The discussion is descriptive and does not compare the data obtained therein with other similar studies except to say that some highly expressed TFs do not yet have associated functions in hindlimb. The overall importance of the findings is not communicated considering other similar published studies.

One minor point related to the manuscript is that the categorization of modules and TF networks using colors in both cases is very confusing. It is unclear whether the reader is meant to associate a module and a network of the same color. If not, then one should be categorized by letters or numbers if the other is by color.  

Reviewer 2 Report

Review of manuscript – Identification of transcription factors during mouse hindlimb development

Although the presentation is acceptable for investigators familiar with WGCNA analysis and other methods, for others, it would be useful to provide a better explanation as to why the approaches were chosen and how they worked. The figures need to be improved as it is impossible to read the small fonts and the Figure legends are only useful to investigators schooled in the approaches. There are no Figure legends for the supplementary Figures and adding these will help. It would be a pity to publish an in-depth study without addressing the above issues and my overall recommendation is to expand the manuscript to improve comprehension for investigators that are unfamiliar with the approaches.

Minor considerations:

Title: network to networks

Abstract: 14 is – are, in terms of seven modules, in fact there are nine and I am not sure why the black and green-yellow were not considered as swing modules.

Keywords: there are only four and adding some of the important transcription factors or specific hindlimb tissues in which arrays are uncovered will attract more readers.

Introduction: 

45 intensive study – gaining understanding

46 is helpful – will provide a basis

51 allow to find – enable investigators to identify hubs

52 why mention differences in hindlimb and forelimb with adding ideas about the differences?

57 an important role – important roles

58 Need to explain why inhibitions of four TFs was performed

Figure 1 – can’t read – need larger fonts and a more detailed legend

65 B – C

105 – by – from

Results:

125 unified – normalized (?)

136 I have no idea how you find nine modules from Figure 2B. Can this Figure be improved, and the colors used in Figure 2C related to lines in Figure 2B. IF I don’t understand, others won’t either and the Figure legend and text in manuscript needs work

139 Should turquoise be brown. This would increase the number from 7 to 8. 

140 Need a better explanation for including Figure 2S and also a legend to the supplement.

Figure 3 – can’t read the words associated with the histograms and what the columns represent

148 I need to understand why black and green-yellow were not considered swing modules.

209 Why were these four transcription factors chosen?

Did any results from this study support roles for PITX1 and/or TBX4 in hindlimb development.

Although the discussion was clear and covered the scope of results, it could be improved by considering how these findings lead to further investigations, further mRNA suppression studies, looking to see if abnormalities in any of the TFs lead to hindlimb abnormalities in humans and how locating mRNAs by in situ hybridization may add support and strengthen results and conclusions.

Round 2

Reviewer 1 Report

The discussion has been improved. However, in reply to a comment about the limitations of this approach compared to RNA-seq analysis, the authors noted "that bulk transcriptome data and transcription factor analysis still gave a lot of useful results". The authors should address this critique in the introduction. 

Reviewer 2 Report

I appreciate the efforts made to address my concerns. I have good eyesight and still feel the fonts in many figures are annoyingly small. 

After making these very minor changes, the paper should be accepted :

Figure 1 – The fonts in B and C are still too small and the colors of yellow and orange are not correct, they look like a light green and red.

Figure 2 – The fonts in A and B are still too small.

Figure 3 – The fonts for the histograms are still too small.

Figure 4 – the transcription factor gene names in the circles are still too small

Line 209 - redder should be redness

Section 3.5 - In choosing the four hub genes, they should be marked in Figure 4B so one can easily locate them.

Line 382 – bule - blue
